# Challenging America: Digitized Newspapers as a Source of Machine Learning Challenges

**Filip Graliński**
Faculty of Math. and Computer Science
Adam Mickiewicz University
filipg@amu.edu.pl

**Jakub Pokrywka**
Faculty of Math. and Computer Science
Adam Mickiewicz University
jakub.pokrywka@amu.edu.pl

**Krzysztof Jassem**
Faculty of Math. and Computer Science
Adam Mickiewicz University
jassem@amu.edu.pl

**Krzysztof Jurkiewicz**
Faculty of Math. and Computer Science
Adam Mickiewicz University
krzysztof.jurkiewicz@amu.edu.pl

**Piotr Wierzchoń**
Faculty of Modern Languages and Literatures
Adam Mickiewicz University
wierzch@amu.edu.pl

**Karol Kaczmarek**
Faculty of Math. and Computer Science
Adam Mickiewicz University

Applica.ai
karol.kaczmarek@applica.ai

## Abstract

This paper introduces an ML challenge, named Challenging America (ChallAm), based on OCR excerpts from historical newspapers collected on the Chronicling America portal. ChallAm provides a dataset of OCR excerpts, labeled with meta-data on their origin and paired with their textual contents retrieved by an OCR tool. Three ML tasks are defined in the challenge: determining the article date, detecting the location of the issue, and deducing a word in a text gap. The challenge is published on the Gonito platform, an evaluation environment for ML tasks, which presents a leader-board of all submitted solutions. Baselines are provided in Gonito for all three tasks of the challenge.

## 1 Introduction

The expansion of digital information is proceeding in two directions on the temporal axis. In the forward direction, new data are made publicly available on the Internet every second. In the backward direction, older and older historical documents are digitized and disseminated publicly. The extraction of relevant information from the overwhelming amount of data is one of the key topics in information processing. This paper concerns the problem of information extraction in the data appearing in the backward temporal direction: OCR historical documents. We use one of the richest sources of such documents, Chronicling America. Based on selected excerpts from Chronicling America we define a new challenge (Challanging America, ChallAm), which comprises three novel ML tasks combining layout recognition, Key Information Extraction (KIE) and semantic inference. We believe that our challenge will contribute to the development of ML methods for the processing of digitized historical resources. We also hope that ChallAm may give rise to a historical equivalent of the GLUE [29] or SuperGLUE [28] benchmarks.

Submitted to the 35th Conference on Neural Information Processing Systems (NeurIPS 2021) Track on Datasets and Benchmarks. Do not distribute.

The paper is organized as follows: In Section 2 we present the Chronicle America website and show how useful this resource may be for humanities research. Section 3 is devoted to existing datasets of historical OCR documents and ML challenges similar to ours. We conclude the section with the statement that there is a need for an ML challenge based on historical OCR texts that goes beyond retrieving explicit information (such as a text fragment or layout components) in the direction of information inference. We propose such a contribution – ChallAm – in subsequent sections. In Section 4 we supply technical details on the processing of supervised data from the Chronicling America corpus. Section 5 is devoted to unsupervised data from the same source, which we release (this is a dump of all Chronicling America texts). In Section 6 we describe the procedure that we applied to prepare the ChallAm challenge. Section 7 describes the three tasks in the challenge: one of them (RetroGap) evaluates the language model directly, while the other two (RetroTemp and RetroGeo) do so indirectly. For each task, we provide several baselines, which are discussed in Section 8. Ethical issues relating to our contribution are considered in Section 9. We conclude the paper with encouragement to define new ML tasks within the ChallAm challenge.

## 2  Chronicling America

In 2005 a partnership between the National Endowment for the Humanities and the Library of Congress launched the National Digital Newspaper Program, to develop a database of digitized documents with easy access. The result of this 15-year effort is Chronicling America – a website[1] which provides access to selected digitized newspapers, published from 1690 to the present. The collection includes approximately 140 000 bibliographic title entries and 600 000 separate library holdings records, converted to the MARCXML format.[2] The portal supports an API which allows accessing of the data in various ways, such as the JSON[3] format, BulkData (bulk access to data) or Linked Data,[4] or searching of the database with the OpenSearch protocol.[5] The newspaper materials in Chronicling America include the following basic elements:

- an uncompressed TIFF representation,
- a compressed JPEG2000 representation,
- PDF with a hidden text layer,
- single-page machine-readable text in column-reading order,
- XML data objects describing newspaper issues, pages, and microfilm reels (metadata).

This five-fold structure of database elements makes Chronicling America a valuable source for the creation of datasets and benchmarks.

The portal serves as a resource for various research activities. Cultural historians may track performances and events of their interest in a resource which is easily and openly accessible, as opposed to commercial databases or "relatively small collections of cultural heritage organizations whose online resources are isolated and difficult to search" [4]. The database enables searching for the first historical usages of word terms. Thanks to the Chronicling America portal, it is discovered in [3] that the term "fake news" was first used in 1889 in the Polish newspaper *Ameryka* (the first use of the term found in an article title is from 1890 in *Daily Tobacco-Leaf Chronicle*). In 2016, research on the etymology of the word "Hoosier"[6] was reported with the statement that thanks to the Chronicling America portal it was possible to obtain "new insights on the term *hoosier*'s usage through time and across geographies."

An interesting case of linguistic research with the aid of the resource is described in [2], where 230 000 pages of historical newspapers from the Chronicling America portal were taken as input data. Using open-source widgets for data visualization, such as Google Maps, Google finance time series, a scrollable timeline of Texas history and a Stanford charting library (applied for plotting word

---

[1] https://chroniclingamerica.loc.gov

[2] The MARC format is a standard for the representation and communication of bibliographic and related information in machine-readable form.

[3] https://www.json.org/json-en.html

[4] https://www.w3.org/standards/semanticweb/data

[5] https://opensearch.org/

[6] https://centerfordigschol.github.io/chroniclinghoosier/

statistics), the research group designed a geo-spatial visualization tool. The user can scroll over time (by means of a slider) and/or location (on a map) to see how historical documents are distributed across both time and space with respect to either quantity and quality of the digitized content or the most widely used large-scale language pattern metrics, such as Word Count, NER Count or Topic Modeling. Topic Modeling itself was examined in [32]. There, a subset of the Chronicling America data, namely digitized newspapers published in Texas from 1829 to 2008, was automatically processed to find changes in topics of interest over time.

Chronicling America may also be of use in prosopography: "an investigation of the common characteristics of a group of people in history, by a collective study of their lives." In [16] a sample from Chronicling America, namely 14,020 articles from a local newspaper, *The Sun*, published in New York in 1896, formed a training set for the task of extracting a people gazetteer,[7] for possible use in prosopographical research.

The resource is helpful in research to improve the output of the OCR process. The authors of [21] study OCR errors occurring in several digital databases – including Chronicling America – and compare them with human-generated misspellings. The research results in several suggestions for the design of OCR post-processing methods. The implementation of an unsupervised approach in the correction of OCR documents is described in [7]. Two million issues from the Chronicling America collection of historic U.S. newspapers are used in a sequence-to-sequence model with attention. This unsupervised model competes with supervised models with respect to character and word error rates.

As shown above, Chronicling America is a type of digitized resource that may be of wide use for humanities research. We prepared datasets and challenges based on the data from the Chronicling America resource. We hope that our initiative will bring about research that will facilitate the development of ML-based processing tools, and consequently increase access to digitized resources for the humanities.

An example of an efficient ML tool based on Chronicling America is described in [18]. The task consisted in predicting bounding boxes around various types of visual content: photographs, illustrations, comics, editorial cartoons, maps, headlines and advertisements. The training set was crowd-sourced and included over 48K bounding boxes for seven classes (among which headlines and advertisements were represented by over 40K classes). Using a pre-trained Faster-RCNN detection object, the researchers achieved an average accuracy of 63.4%. Both the training set and the model weights file are publicly available. Still, it is difficult to estimate the value of the results achieved without any comparison with other models trained on the same data.

In our proposal we go a step further. We provide and make available training data from Chronicling America for three ML tasks. For each task we develop and share baseline solutions. Moreover, we share a platform that supports the evaluation of competing solutions for each task, called Gonito [11]. Any competitor may upload their contribution to Gonito (`https://gonito.net`), and the system automatically compares its performance with the baseline and other uploaded solutions.

## 3   Similar Machine Learning datasets and challenges

### 3.1   Datasets

The datasets of our interest are collections of OCR newspapers with metadata that may be used in supervised learning.

In [5] a ground-truth[8] dataset of European historical newspapers is described. The dataset comprises over 500 pages representing 12 European languages. Each page is labeled with full text in Unicode (with reading order), precise region outlines, and region type labels, such as Regions (blocks/zones), Images/Graphics, Tables, Text Regions or Text Lines.

The Dutch laboratory KB Lab offers a collection of datasets containing historical newspapers. The Historical Newspaper OCR Ground Truth[9] set offers 2 000 pages (one page per newspaper issue) from historical sources. Each JPG2 image of a page is accompanied by the outcome of OCR processing as

---

[7]A people gazetteer consists of personal names along with lists of documents in which they occur.

[8]The term "ground-truth" refers to a perfect (usually manually verified) outcome of OCR processing.

[9]https://lab.kb.nl/dataset/historical-newspapers-ocr-ground-truth

well as manually corrected ground-truth text files. The SIAMESET dataset [30] contains over 450K advertisements in the form of JPEG images from two Dutch newspapers dated from 1945 to 1994. The dataset provides metadata for each of the images (date, size, position, page number, etc.) and the textual content recognized by the OCR software. A huge dataset (102 million news items) from Dutch historical newspapers is described in [22]. Combining the Named Entity Recognition techniques and disambiguation algorithms, the authors of the dataset succeeded in marking occurrences of city names in the news items. The CHRONIC dataset [27] consists of 313K classified images from Dutch digitized newspapers for the period 1860–1922. The images are automatically classified into one of nine categories: buildings, cartoons, chess, crowds, logos, maps, schematics, sheet music and weather reports.

American digitized documents are preserved by the UNT (University of North Texas) Digital Library. The library provides, among other digitized resources, a dataset of OCR texts from two Houston newspapers from the years 1893 to 1924 [24]. The dataset includes 184 900 pages of text.

## 3.2 ML challenges

This section concerns ML challenges which deliver labeled OCR documents as training data, a definition of the processing task, and an evaluation environment to estimate the performance of uploaded solutions. More often than not, such challenges concern either layout recognition (localization of layout elements) or Key Information Extraction (finding, in a document, precisely specified business-actionable pieces of information). Layout recognition in Japanese historical texts is described in [26]. The authors use deep learning-based approaches to detect seven types of layout element categories: Page Frame, Text Region, Text Row, Title Region, Title, Subtitle, Other. Some Key Information Extraction tasks are presented in [15]. The two datasets described there contain, respectively, NDA documents and financial reports from charity organizations, all in the English language. The tasks for the first dataset consist in the detection of effective dates, interested parties, jurisdiction, and terms. The tasks for the second dataset consist in the recognition of towns, postcodes, streets, charity names, charity numbers, income, report dates and spending. The authors provide several baseline solutions for the two tasks, which apply up-to-date methods, pointing out that there is still room for improvement in the KIE research area. A challenge that comprises both layout recognition and KIE is presented in [17] – the challenge is opened for the recognition of OCR-scanned receipts. In this competition (named ICDAR2019) three tasks are set up: Scanned Receipt Text Localization, Scanned Receipt OCR, and Key Information Extraction from Scanned Receipts.

A common feature of the above-mentioned challenges is the goal of retrieving information that is explicit in the data (a text fragment or layout coordinates). Our tasks in ChallAm go a step further: the goal is to infer the information from the OCR image rather than just retrieve it.

Similar challenges for two out of the three tasks introduced in this paper have been proposed before for the Polish language:

- a challenge for temporal identification [10]; the challenge was based on a set of texts coming from Polish digital libraries, dated between the years 1814 and 2013;
- a challenge for "filling the gap" (RetroGap) [13] with the same training set as above.

The training sets for those challenges were purely textual. Here, we introduce the challenges with the addition of OCR images.

## 4 Data processing

The PDF files were downloaded from Chronicling America and processed using a pipeline primarily developed for extracting texts from Polish digital libraries [12, 14]. Firstly, the metadata (including URL addresses for PDF files) were extracted by a custom web crawler and then normalized; for instance, titles were normalized using regular expressions (e.g. *The Bismarck tribune. [volume], May 31, 1921* was normalized to *THE BISMARCK TRIBUNE*). Secondly, the PDF files were downloaded and the English texts were processed into DjVu files (as this is the target format for the pipeline) using the pdf2dvju tool[10]. The original OCR text layer was retained (the files were not re-OCRed, even though, in some cases, the quality of OCR was low).

---

[10]http://jwilk.net/software/pdf2djvu

Table 1: Statistics for the raw data obtained from the Chronicling America website

| | |
|---|---|
| Documents for which metadata were obtained | 1 877 363 |
| . . . in English | 1 705 008 |
| . . . downloaded | 1 683 836 |
| . . . processed into DjVus | 1 665 093 |

Table 1 shows a summary of the data obtained at each processing step. Two factors were responsible for the fact that not 100% of files were retained at each phase: (1) issues in the processing procedures (e.g. download failures due to random network problems or errors in the PDF-to-DjVu procedure that might be handled later); (2) some files are simply yet to be finally processed in the ongoing procedure of data collection.

The procedure is executed in a continuous manner to allow the future processing of new files that are yet to be digitized and made public by the Chronicling America initiative. This solution requires a *future-proof* procedure for splitting and preparing data for machine-learning challenges. For instance, the assignment of documents to the training, development and test sets should not change when the raw data set is expanded. The procedure is described in Section 6.

## 5   Data for unsupervised training

The state of the art in most NLP tasks is obtained by training a neural-network language model on a large collection of texts in an unsupervised manner and fine-tuning the model on a given downstream task. At present, the most popular architectures for language models are Transformer [6] models (earlier, e.g. LSTM [23] or Word2vec models [20]). The data on which such models are trained are almost always modern Internet texts. The high volume of texts available at Chronicling America, on the other hand, makes it possible to train large Transformer models for historical texts.

Using a pre-trained language model on a downstream task bears the risk of *data contamination* – the model might have been trained on the task test set and this might give it an unfair edge (see [1] for a study of data contamination in the case of the GPT-3 model when used for popular English NLP test sets). This issue should be taken into account from the very beginning. In our case, we release a dump of all Chronicling America texts (for pre-training language models), but limited only to the 50% of texts that would be assigned to the training set (according to the MD5 hash). This dump contains *all* the texts, not just the excerpts described in Section 6.2. As the size of the dump is 171G characters, it is on par with the text material used to train, for instance, the GPT-2 model.

## 6   Procedure for preparing challenges

We created a pipeline that can generate various machine learning challenges. The pipeline input should consist of DjVu image files, text (OCR image), and metadata. Our main goals are to keep a clear distinction between dataset splits and to assure the reproducibility of the pipeline. This allows potential improvement to current challenges and the generation of new challenges without dataset leaks in the future. We achieved this by employing *stable* pseudo-randomness by calculating an MD5 hash on a given ID and taking the modulo remainder from integers from certain preset intervals. These pseudo-random assignments are not dependent on any library, platform, or programming language (using a fixed seed for the pseudo-random generator might not give the same guarantees as using MD5 hashes), so they are easy to reproduce.

This procedure is crucial to make sure that challenges are *future-proof*, i.e.:

- when the challenges are re-generated on the same Chronicling America files, exactly the same results are obtained (including text and image excerpts; see section 6.2);

- when the challenges are re-generated on a larger set of files (e.g. when new files are digitized for the Chronicling America project), the assignments of existing items to the train/dev/test sets will not change.

## 6.1 Dataset structure

All three of our machine learning challenges consist of training (train), development (dev), and test sets. Each document in each set consists of excerpts from a newspaper edition. One newspaper edition provides a maximum of one excerpt. Excerpts in the datasets are available as both a cropped PNG file from the newspaper scan (a "clipping") and its OCR text. This makes it possible to employ image features in machine learning models (e.g. font features, paper quality). A solution might even disregard the existing OCR text layer and re-OCR the clipping or just employ an end-to-end model. (The OCR layer is given as it is, with no manual correction done – this is to simulate realistic conditions in which a downstream task is to be performed without a perfect text layer.)

Sometimes additional metadata are given. For the train and dev datasets, we provide the expected data. For the test dataset, the expected data are not released. These data are used by the Gonito platform during submission evaluation. All newspaper and edition IDs are encoded to prevent participants from checking the newspaper edition in the Chronicling America database. The train and dev data may consist of all documents which meet our text excerpts criteria, so the data may be unbalanced with respect to publishing years. We tried to balance the test sets as regards the years of publication (though it is not always possible due to large imbalances in the original material).

## 6.2 Selecting text excerpts

The OCR text follows the newspaper layout, which is defined by the following entities: page, column, line. Each entity has $x_0, y_0, x_1, y_1$ coordinates of text in the djvu document. Still, various errors may occur in the OCR newspaper layout (e.g. two columns may be split into one). We intend to select only excerpts which preserve the correct output. To this end, we select only excerpts that fulfill the following conditions:

1. There are between 150 and 600 text tokens in the excerpt. The tokens are words separated by whitespaces.

2. The $y$ coordinates of each line are below the $y$ coordinates of the previous line.

3. The $x_0$ coordinate of each line does not differ by more than 15% from the $x_0$ coordinate of the previous line.

4. The $x_1$ coordinate is not shifted to the right more than 15% from the $x_1$ coordinate of the previous line.

If the newspaper edition contains no such excerpts, we reject it. If there is more than one such excerpt, we select one excerpt using a stable pseudo-random procedure based on the newspaper edition ID (as described earlier).

This procedure produces text excerpts with images consisting of OCR texts only. The excerpts are downsized to reduce the size to an appropriate degree to maintain good quality. We do not preprocess images in any other way, so excerpts may have different sizes, height-to-width ratios, and colors. A sample excerpt is shown in Figure 1a.

## 6.3 Train/dev/test split

Each newspaper has its newspaper ID, and each newspaper edition has its newspaper edition ID. We separate newspapers from datasets, so for instance, if one newspaper edition is assigned to the dev set, all editions of that newspaper are assigned to the dev set. All challenges share common train and dev datasets and no challenges share the same test set. This prevents one from checking expected data from other challenges. The set splits are as follows:

- 50% for train;

- 10% percent for dev;

- 5% percent for each challenge test set.

This makes it possible to generate eight challenges with different test sets. In other words, there is room for another five challenges in the future (again this is consistent with the "future-proof" principle of the whole endeavor).

Perhaps one of the most interesting
political developments in tbe political
history of California is that which has
been disclosed as a result of the quarrel
of Leland Stanford and Collis P. Hunt-
ington, of the Southern and Central Pa-
cific Railways, and which has been sup-
pressed as to details, after the scandal
has embraced a whole continent. It is
probable that much matter for good will
ultimately result from this and other
indecent developments. Prior to the ar-
rival of Mr. Huntington on this Coast
the people of California were in danger
of being deluged in a stream of adula-
tion directed towards Senator Stanford.
Although Stanford notoriously pur-
chased his seat in the United States
Senate, and although bis purchase of
that seat, considering his obligations to
Senator Sargent, was a matter of never
to be forgottoa treachery, the toad-
eaters of the might}' Senator are
intent upon having censers swung in his
...

(a) An excerpt.          (b) Fragment of a text from an excerpt.

## 7  Challenging America tasks

In this section, we describe the three tasks defined in the challenge. They are released on the Gonito platform, which enables the calculation of metrics both offline and online, as well as the submission of solutions and tracking of leader-boards. An example of text from an excerpt given in those tasks is shown in Figure 1b.

### 7.1  RetroTemp

This[11] is a temporal classification task. Given a normalized newspaper title and a text excerpt, the task is to predict the publishing date of the newspaper edition. The date should be given in fractional year format (e.g. 1 June 1918 is represented as the number 1918.4137, and 31 December 1870 as 1870.9973), which can be described by the following code:

```
fractional_day = (60*60*hour+60*minute+second) / (24*60*60)
days_in_the_year = 366 if year_is_leap_year else 365
fractional_year = year + (day_in_year-1+fractional_day) / days_in_the_year
```

Hence, solutions to the challenge should predict the publication date with the greatest precision possible (i.e. day if possible). The fractional format will make it easy to accommodate even more precise timestamps, for example, if modern Internet texts (e.g. tweets) are to be added to the dataset.

Due to the regression nature of the problem, the evaluation metric is RMSE (root mean square error).

The motivation behind the RetroTemp challenge is to design tools that may help supplement the missing metadata for historical texts (the older the document, the more often it is not labeled with a time stamp). Even if all documents in a collection are time-stamped, such tools may be useful for finding errors and anomalies in metadata.

---

[11]https://gonito.net/challenge/challenging-america-year-prediction

## 7.2 RetroGeo

The task[12] is to predict the place where the newspaper was published, given a normalized newspaper title, text excerpt, and publishing date in fractional year format. The expected format is a latitude and longitude. In the evaluation the distance on the sphere between output and expected data is calculated using the haversine formula, and the mean value of errors is reported.

The motivation for the task (besides the supplementation of missing data) is to allow research on news propagation. Even if a news article is labeled with the localization of its issue, an automatic tool may infer that it was originally published somewhere else.

## 7.3 RetroGap

This[13] is a task for language modeling. The middle word of an excerpt is removed in the input document (in both text and image), and the task is to predict the removed word, given the normalized newspaper title, the text excerpt, and the publishing date in fractional year format. The output should contain a probability distribution for the removed word in the form:

$$word_1 : logprob_1 \; word_2 : logprob_2 \; \ldots \; word_N : logprob_N : logprob_0$$

where $logprob_i$ is the logarithm of the probability for $word_i$ and $logprob_0$ is the logarithm of the probability mass for all other words. It is up to the submitter to choose the words and $N$. The metric is LikelihoodHashed as introduced in [13], to ensure proper evaluation in the competitive (shared-task) setup.

A standard metric for language model evaluation is perplexity, which tells how much the probability distributions on word sequences predicted by the model differ from distributions in the test set – the lower the perplexity, the better the model. The downside of the perplexity metric is that the its reported values are difficult to verify. Recent findings ([8]) point out the need for defining new intrinsic method of language evaluation. ChallAm follows this direction – it provides a task for predicting a word in a given context for the evaluation of the models that represent a historical language. The metric used in the challenge is LikelihoodHashed, which allows for objective comparison of all reported solutions.

## 7.4 Statistics

The data consists of the text excerpts written between the years 1798 and 1963. The mean publication year of the text excerpts is 1891. Excerpts between the years 1833 and 1925 make up about 97% of the data in the train set (cf. Figure 2a), but only 88% in the dev and test sets, which are more uniform (cf. Figure 2c).

There are 398 800 excerpts in the train set, 9 400 in the dev set and 9 500 in the test set. These numbers are consistent across the challenges. The average excerpt length is 1 745 characters with 322.8 words, each one contain from 150 words up to 559 words.

The length of each text in the excerpts seems to have a negative correlation with publication date – the later the text was published, the shorter snippet text (on average) it contains (see Figure 2b and Figure 2d).

## 8 Baselines

Baselines for all three tasks are available at `https://gonito.net`. The baselines (see Tables 2 and 3) include, for each model, its score in the appropriate metric as well as the Git SHA1 reference code in the Gonito benchmark (in curly brackets). Reference codes can be used to access any of the baseline solutions at `http://gonito.net/q`. The baseline source codes are provided in corresponding repositories.

We distinguish between self-contained submissions, which use only data provided in the task, and non-self-contained submissions, which use external data, e.g. publicly available pre-trained transformers. Our baselines take into account only textual features.

---

[12]`https://gonito.net/challenge/challenging-america-geo-prediction`
[13]`https://gonito.net/challenge/challenging-america-word-gap-prediction`

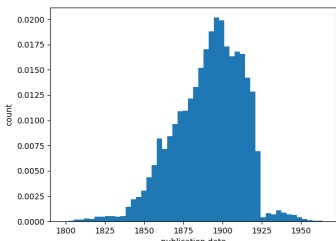

(a) Excerpt counts vs. publication dates in train set.

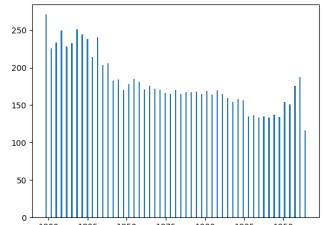

(b) Average excerpt length vs. publication dates in train set.

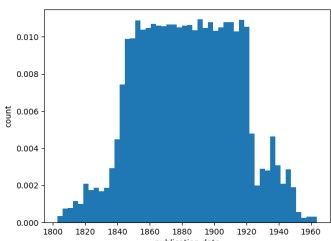

(c) Excerpt counts vs. publication dates in dev/test set.

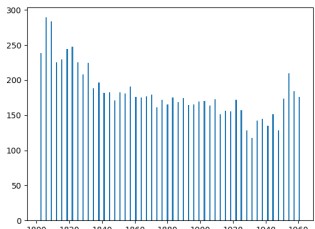

(d) Average excerpt length vs. publication dates in dev/test set.

Table 2: Baseline results for the RetroTemp/Geo challenges. * indicates non-self-contained models.

| Model | RetroTemp | | RetroGeo | |
| --- | --- | --- | --- | --- |
| | Gonito ref | RMSE | Gonito ref | Haversine |
| mean from train | {fbf19b} | 31.50 | {766824} | 1321.47 |
| tf-idf with linear regression | {63c8d4} | 17.11 | {8acd61} | 2199.36 |
| BiLSTM | {f7d7ed} | 13.95 | {d3d376} | 972.71 |
| RoBERTa base with linear layer* | {611cbc} | 10.19 | {08412c} | 827.13 |
| RoBERTa large with linear layer* | {2e79c8} | **8.15** | {7a21dc} | **651.20** |

## 8.1 RetroTemp and RetroGeo

The baseline solutions for RetroTemp and RetroGeo were prepared similarly. RetroGeo requires two values (latitude and longitude) – we treat them separately and train two separate models for them.

For the self-contained models we provide the mean value from the train test, the linear regression based on TF-IDF and the BiLSTM (bidirectional long short-term memory) method.

For non-self-contained submissions, we incorporate RoBERTa [19] models released in two versions: base (125M parameters) and large (355M parameters). The output features are averaged, and the linear layer is added on top of this. Both RoBERTa and the linear layer were fine-tuned during training.

The best self-contained models are BiLSTM submissions in both tasks. Non-self-contained submissions result in much higher scores than self-contained models. In both tasks, RoBERTa large with linear layer provides better results than RoBERTa base.

## 8.2 RetroGap

All RetroGap baselines do not employ the publishing year as a feature. For non-self-contained submissions, we trained the BiLSTM and Transformer models in two ways: using vocabulary based on single words (word models) and based on the BPE [25] subwords (BPE models). For self-contained submissions, we applied RoBERTa with and without fine-tuning.

Table 3: Baseline results for the RetroGap challenge. * indicates non-self-contained models.

| Model | Gonito ref | LikelihoodHashed |
|---|---|---|
| BiLSTM (word) | {ae5c5e} | 0.00565 |
| Transformer (word) | {41e8ce} | 0.00399 |
| BiLSTM (BPE) | {4445ca} | 0.00558 |
| Transformer (BPE) | {941993} | 0.00539 |
| RoBERTa base (no finetune)* | {43ecf5} | 0.01534 |
| RoBERTa base (finetune)* | {2b3951} | 0.01878 |
| RoBERTa large (no finetune)* | {bf5171} | 0.02006 |
| RoBERTa large (finetune)* | {4ba590} | **0.02186** |

Again, non-self-contained models achieve much better scores than self-contained models. The best pre-trained model is RoBERTa large fine-tuned to the task data. Among self-contained models, BiLSTM achieves better results than Transformer. The best self-contained model is BiLSTM based on word tokenization.

## 9   Ethical issues

We share the data from Chronicling America, following the statement of the Library of Congress: "The Library of Congress believes that the newspapers in Chronicling America are in the public domain or have no known copyright restrictions."[14]

The data sets are provided "as is" and reflect views and opinions from their period of origin. We are aware of the fact that historical texts from American newspapers may be discriminatory, either explicitly or implicitly, particularly regarding gender and race. Recent years have seen research on the detection of discriminatory texts. In [31] adversarial training is used to mitigate racial bias. In [9] the authors "take an unsupervised approach to identifying gender bias against women at a comment level and present a model that can surface text likely to contain bias." Most research on the topic concerns contemporary texts. ChallAm provides the opportunity for similar investigation of historical texts. A model trained on historically accurate data could potentially be used to detect and correct discriminatory texts.

## 10   Conclusions

This paper has introduced a challenge based on OCR excerpts from the Chronicling America portal. The challenge consists of three tasks: guessing the publication date, guessing the publication location, and filling a gap with a word. We propose baseline solutions for all three tasks.

Chronicling America is an ongoing project that is very useful for humanities research. We define our challenge in such a way that it can easily evolve in parallel with the development of Chronicling America. Firstly, any new materials appearing on the portal can be automatically incorporated into our challenge. Secondly, the challenge is open for new yet undefined ML tasks.

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
