# OpenReview forum: "Challenging America: Digitized Newspapers as a Source of Machine Learning Challenges"
_NeurIPS.cc/2021/Track/Datasets_and_Benchmarks/Round1 — Submitted to NeurIPS 2021 Datasets and Benchmarks Track (Round 1)_

### Official Review · Reviewer_txz2 · 2021-07-01
**Less innovative machine learning challenges for digitized news**

**Rating:** 7
**Confidence:** 3
**Correctness:** Yes.
**Clarity:** Yes.

**Strengths:**

- Interesting and valuable historical news dataset.
- Sound and correct scientific procedure.
- Both code and data are available and there seem to be no ethical questions.


**Weaknesses:**

- Too much emphasis in the ML tasks and no clear motivation for those.
- Missing to show more information on the dataset.



**Additional Feedback:**

This paper gives more importance to the tasks and less to the dataset. For instance, little attention is given to the available variables in the dataset, (e.g., it would be interesting to have more details on dataset descriptives).

**Documentation:**

Yes.

**Ethics:**

No.

**Relation To Prior Work:**

Yes.

**Summary And Contributions:**

This paper presents a new machine learning (ML) task in Gonito platform where the goal is to analyze digitized newspapers. The data consists of Chronicling America news written between the years 1798 and 1963 in image format. Excerpts between the years 1833 and 1925 make about 97 % of all data.

Comparison with previous datasets is provided. Similar machine learning tasks are also briefly discussed which gives a proper related work section to the paper.

The proposed ML tasks are simple and not very innovative:
- RetroTemp, given a normalized newspaper title and a text excerpt, the task is to predict the publishing date of the newspaper edition.
- RetroGeo, given the normalized newspaper title, text excerpt, and publishing date, the task is to retrieve the place where the newspaper is published (latitude and longitude).
- RetroGap, given the normalized newspaper title, the text excerpt, and the publishing date. The middle word of an excerpt is removed in the input document (in both text and image), and the task is to predict the removed word.

Undoubtedly the dataset with historical news is relevant, however, the paper could better motivate the need for such tasks. This is even more evident for the RetroGap task which in my opinion has little applicability.

Regarding data divisions for train/dev/test split, the work separates newspapers from datasets, so for instance, if one newspaper edition is assigned to the dev set, all editions of that newspaper are assigned to the dev set. This procedure increases the dataset quality and decreases the risk of training newspaper-specific overfitted models.

Finally, the paper provides baselines for the tasks where BiLSTM was trained and Transformer models were used.

This paper gives more importance to the tasks and less to the dataset. For instance, little attention is given to the available variables in the dataset, (e.g., it would be interesting to have more details on dataset descriptives).

---

> ### Author Response · Authors · 2021-07-12
> **Resubmission with improved motivation, baselines, statistics.**
>
> Thank you for your valuable review! To address the remarks of the reviewers, we have re-submitted a revised version of our paper.
>
> In the Introduction of the resubmitted version we discuss the motivation for all three tasks in general: they may serve for either direct (RetroGap) or indirect (RetroTemp, RetroGeo) evaluation of a historical language model. Besides, the motivation for each separate task is presented in Section 7 of the revised version: RetroTemp and RetroGeo may help train tools for the supplementation of missing metadata, the correction of errors in the existing metadata, and anomaly detection. Moreover, RetroGeo may allow research on news propagation. Even if a news article is labeled with the localization of its issue, an automatic tool may infer that it was originally published somewhere else. RetroGap is a standard challenge for the evaluation of language models -- defined here for historical texts.
>
> In the revised version, we discuss the baselines more deeply in Section 8. We provide 5 baselines for RetroTemp, 5 baselines for RetroGeo and 8 baselines for RetroGap. For each baseline, we describe the applied methodology, present its score in the appropriate metric and provide the Gonito reference code. The reference codes can be used to access any of the source codes for baseline solutions.
>
> In the revised version we present more details on the statistics of our dataset. Section 7.4 now contains more numerical information as well as new figures that illustrate the statistics.

---

> > ### Comment · Reviewer_txz2 · 2021-07-14
> > **Raised review from 6 to 7.**
> >
> > I thank the authors for the changes in the paper and the detailed description of those.
> > Regarding the motivation for the tasks, 7.1 and 7.2 are now more convincing, but I am still not optimistic regarding the pertinence of task 7.3.
> >
> > The baselines look now better and also the additions on section 7.4.
> >
> > Given the new details, I am positive to raise my review in one point (from 6 to 7).

---

> > > ### Author Response · Authors · 2021-07-14
> > > **subsection 7.3 improved**
> > >
> > > Thank you very much for your positive response to our re-submition! The new remarks are very valuable for us, particularly the critical one: ''but I am still not optimistic regarding the pertinence of task 7.3.''
> > >
> > > Other reviewers have pointed out the same drawback. Therefore we have decided to upload yet another submission, which more deeply presents the motivation for the task in subsection 7.3.

---

### Official Review · Reviewer_kqen · 2021-07-04
**Interesting OCR++ dataset**

**Rating:** 6
**Confidence:** 2

**Strengths:**

- the dataset motivation (Line 39-85) and the comparison (Line 96-116) with existing work is really well done. These two paragraphs explain in depth the importance of this dataset. The pre-processing step in Section 6.2 is quite clear as well.
- Moreover, since news article spans from 1798 to 1963, can be a very useful resource to train LM as well, and especially to compare the representation of text through time.


**Weaknesses:**

- the paper structure is a bit verbose and misses some important aspect as Train/Val/Test samples etc. Although the paper is well-written, it is very hard to grasp the main idea and the dataset tasks until page 5 or 6, which makes the task hard to understand. I would suggest moving the related work to a specific section and focus first on the proposed dataset and task. Moreover, many important statistics are not mentioned in the paper, such exact number of samples for each task, and misses examples for each of the task.
- missing baselines for the RetroGeo and RetroTemp in the paper. The results are reported in the leaderboard, but there I don't know if these number are from the authors or from some other participants. Without adding this baseline is hard to evaluate how hard these tasks are.  Moreover, it is not clear what are the performance of an E2E system where the input is the img directly.

**Additional Feedback:**

- Adding a table that compares existing dataset statistics would be very useful for the readers

**Clarity:**

Overall, the paper is clearly written. However, the structure of the paper needs work especially, it does not get to the point until pages 5 o 6.

**Correctness:**

Yes, the dataset seems contructed in a sound way. However the experimental details and design are very poor, missing many baselines of the proposed tasks.

**Documentation:**

The author posted the data on a platform and based on the information in the paper, this dataset is highly reproducible. To be honest, I navigate a bit the hosted platform with the data (https://gonito.net/challenge/challenging-america-year-prediction)  and it is very hard to navigate. After checking around,  and many clicks, I could find a way to download the data.

**Ethics:**

No particular.

**Relation To Prior Work:**

The author did an excellent job here.

**Summary And Contributions:**

In this paper, the authors present a new OCR excerpts dataset for historical newspapers from Chronicling America. The dataset is made of 1.6M documents crawled and pre-processed from Chronicling America, and it includes three subtasks: RetroTemp (the task of predicting the publishing date of the article given the title, and excerpts), RetroGeo (the task of predicting latitude and longitude given title, excerpt, and publishing date), and RetroGap (the task of predicting a masked word -- both in the text and the image -- in the context article). The authors report some preliminary results on the RetroGap and several statistics on the collected dataset.

---

> ### Author Response · Authors · 2021-07-12
> **Resubmission with improved introduction, statistics, and added baselines.**
>
> Thank you for your valuable review! To address the remarks of the reviewers, we have re-submitted a revised version of our paper.
>
> To help the reader grasp the idea as soon as possible, we have re-written the Introduction section. In the current version, we immediately get to the point: the Introduction describes the framework of our challenge and presents the general motivation for the tasks (direct and indirect evaluation of historical language models). The current introduction overviews the structure of the paper, hopefully clarifying why we decide to cite related work prior to introducing our contribution. More specifically, in Section 3 we aim to demonstrate the importance of Chronicling America for humanities research, and in Section 4 we conclude that there is a need for an ML challenge, Iike ours, based on historical OCR texts that goes beyond retrieving explicit information.
>
> In the revised version we present more details on the statistics of our dataset. Section 7.4 now contains more numerical information as well as new figures that illustrate the statistics.
>
> In the revised version, we discuss the baselines more deeply in Section 8. We provide 5 baselines for RetroTemp, 5 baselines for RetroGeo and 8 baselines for RetroGap. For each baseline, we describe the applied methodology, present its score in the appropriate metric and provide the Gonito reference code. The reference codes can be used to access any of the source codes for baseline solutions. We also clarify that all baselines are based only on textual features (we do not provide E2E baselines with img input).

---

> > ### Comment · Reviewer_kqen · 2021-07-15
> > **Raise from 5 to 6**
> >
> > Thank you for updating your paper.  My main concerns are addressed.

---

### Official Review · Reviewer_5KFt · 2021-07-04
**Doubtful Utility with Potential Ethical Problems**

**Rating:** 5
**Confidence:** 3
**Correctness:** The dataset is constructed in a reaso…

**Strengths:**

- This paper introduces a new large-scale dataset, which may be beneficial for humanities research.
- The large-scaled dataset may be beneficial for training language models.

**Weaknesses:**

- I don't really see the utility of the introduced three tasks. Why people should work on the dataset is unclear.
- There is a lack of ethical discussions for the dataset, while there's a high chance that social biases are contained. As a result, models trained on this data may also be discriminatory.
- The paper is poorly presented.


**Additional Feedback:**

N/A

**Clarity:**

The presentation of the paper is weak. The formats of the figures (e.g. Figure 1) and tables (e.g. Table 2) are weird. For wrap figures, you can use vspace{-xmm} to adjust the top and bottom spacing. Also, I don't understand why the Baseline results of RetroTemp and RetroGeo are not presented in the paper as there are still a lot of spaces left.

**Documentation:**

- The authors provide the details for the data preparing process and made an effort for reproducibility.
- Ethical concerns are not discussed in this work.


**Ethics:**

The paper did not discuss the potential ethical problems in the dataset. Considering that it's collected from the news in past years from 1798, there's a great chance that it contains discriminatory texts in the dataset. The authors should make further verification, processing, and discussion for this issue. This problem also makes training LM on this dataset dangerous.

**Relation To Prior Work:**

The paper has clearly discussed its relation with previous works.

**Summary And Contributions:**

The paper provides a dataset named ChallAm, which is collected from historical newspapers collected on the Chronicling America portal. The resulted dataset contains 1665093 OCR image - text pairs Correspondingly, three tasks are introduced, including news date and location detection and language modeling. The challenge is published on a publicly available platform with a leaderboard.

---

> ### Author Response · Authors · 2021-07-12
> **Resubmission with improved motivation, formatting, added baselines and Ethical issues sections.**
>
> Thank you for your valuable review! To address the remarks of the reviewers, we have re-submitted a revised version of our paper.
>
> In the Introduction of the resubmitted version we discuss the motivation for all three tasks in general: they may serve for either direct (RetroGap) or indirect (RetroTemp, RetroGeo) evaluation of a historical language model. Besides, the motivation for each separate task is presented in Section 7 of the revised version: RetroTemp and RetroGeo may help train tools for the supplementation of missing metadata, the correction of errors in the existing metadata, and anomaly detection. Moreover, RetroGeo may allow research on news propagation. Even if a news article is labeled with the localization of its issue, an automatic tool may infer that it was originally published somewhere else. RetroGap is a standard challenge for the evaluation of language models -- defined here for historical texts.
>
> We have added a new section entitled ‘‘Ethical issues”. There, we discuss the copyright restrictions on texts from Chronicling America. We also address the problem of discriminatory bias in historical texts. We conclude the section by expressing the hope that the ChallAm challenge may help to train a model that could be used to detect and correct discriminatory texts.
>
> To improve the presentation of the paper, we have substantially modified the Introduction section. We hope that this change will improve the clarity of the paper.
>
> We have made adjustments to the formatting of the paper, as indicated by the reviewer.
>
> In the revised version, we discuss the baselines more deeply in Section 8. We provide 5 baselines for RetroTemp, 5 baselines for RetroGeo and 8 baselines for RetroGap. For each baseline, we describe the applied methodology, present its score in the appropriate metric and provide the Gonito reference code. The reference codes can be used to access any of the source codes for baseline solutions.

---

> > ### Comment · Reviewer_5KFt · 2021-07-15
> > **Improved Motivation and Presentation with Potential Ethical Concerns**
> >
> > The motivation of 7.1 and 7.2 makes more sense now to me, and I appreciate that. The presentation of the paper is also better.
> >
> > For potential ethical problems, the authors add a section to discuss. However, I do not see how this data could be used to "*detect and correct the discriminations*" as no annotations about the biases are included. Actually, I don't see any modifications to the data. Besides, it's mentioned in the Section 5 that the data can be used for training pre-trained language models, but the models trained from this data will be highly likely to contain biases. To address the ethical concerns, for example, the authors could somehow detect the potential bias samples and make an annotation.
> >
> > In sum, I am lean to keep my score unchanged currently until the authors make further updates.

---

### Official Review · Reviewer_h5Ym · 2021-07-04
**A dataset and infrastructure for identifying dates, locations, and missing words in historical newspapers**

**Rating:** 7
**Confidence:** 3

**Strengths:**

The dataset is clearly described and the development of an accompanying platform makes it accessible to other interested researchers. This dataset would likely be useful for researchers interested in classifying unknown historical documents and working on methods to spatiotemporally locate documents based on textual and other visual information. The dataset may also be useful for training language models on historical corpora, although other such resources like Google Ngrams already exist.

**Weaknesses:**



There is little discussion of the motivation for these three tasks in particular. Under what circumstances might one need to identify the date or location of a particular newspaper article, given that such materials are generally accompanied by relevant metadata.

The authors only show results for baselines on one of the tasks. It would be useful to see the results for the other two tasks. I would also appreciate more discussion of the kinds of performance we might expect from such systems. In other words, can we actually predict the date or location of a newspaper article with any reasonable degree of accuracy? To what extent do such predictions depend upon the presence of features like dates, place names, or specific events?

**Additional Feedback:**

N/A

**Clarity:**

The paper is generally well-written. I would recommend making the front-end (literature review) more concise and spending some more time discussing the benchmarks and results.

**Correctness:**

The approach appears to be sound. The authors clearly describe the procedure for selecting documents and take necessary measures to prevent downstream problems (e.g. splitting test/train/dev by publication). It is difficult to evaluate the correctness of the baseline methods as they are only very briefly discussed and the results are not shown.

**Documentation:**

The process is well documented and the dataset is hosted on an open-source platform designed to host ML challenges.

**Relation To Prior Work:**

Yes

**Summary And Contributions:**

This paper introduces a corpus of OCR documents derived from historical newspapers in the United States. The data itself is extracted from the Chronicling America portal and have been processed to make them suitable for the task. The tasks involve viewing a snippet of a given newspaper and variously determining the date of publication, the location (lat/lon), and identifying a missing word. The dataset is designed to serve as a resource for improving ML methods for analyzing historical texts.

---

> ### Author Response · Authors · 2021-07-12
> **Resubmission with improved motivation and baselines.**
>
> Thank you for your valuable review! To address the remarks of the reviewers, we have re-submitted a revised version of our paper.
>
> In the Introduction of the resubmitted version we discuss the motivation for all three tasks in general: they may serve for either direct (RetroGap) or indirect (RetroTemp, RetroGeo) evaluation of a historical language model. Besides, the motivation for each separate task is presented in Section 7 of the revised version: RetroTemp and RetroGeo may help train tools for the supplementation of missing metadata, the correction of errors in the existing metadata, and anomaly detection. Moreover, RetroGeo may allow research on news propagation. Even if a news article is labeled with the localization of its issue, an automatic tool may infer that it was originally published somewhere else. RetroGap is a standard challenge for the evaluation of language models -- defined here for historical texts.
>
> In the revised version, we discuss the baselines more deeply in Section 8. We provide 5 baselines for RetroTemp, 5 baselines for RetroGeo and 8 baselines for RetroGap. For each baseline, we describe the applied methodology, present its score in the appropriate metric and provide the Gonito reference code. The reference codes can be used to access any of the source codes for baseline solutions.

---

### Decision · Program_Chairs · 2021-07-27

**Decision:**

Reject

**Comment:**

This paper introduces three tasks based on a time-stamped, location-stamped large-scale corpus (roughly 398K excerpts for training, 18K for dev/eval), based on OCRed newspapers from the Chronicling America portal. Along with it, it proposes three tasks -- predicting timestamp of text snippet (RetroTemp), predicting the location of the text snippet (RetroGeo), and predicting masked token from the text snippet (RetroGap).

All reviewers (and AC) point that the motivations behind such tasks are rather weak. Further, the paper does not provide detailed documentation of the datasets, and what kind of machine learning challenges are posed by each task.  For example, how good is the OCR performance? what is the bottleneck for current models? are the performances better on newer documents vs. older documents? Specifically, the last task (RetroGap) is very similar to the standard language modeling task, and the authors introduce a less used metric for the last task, LikelihoodHashed, instead of perplexity, without a convincing justification.

Also, while acknowledging potential ethical issues behind the dataset (section 9), the authors do not investigate this quantitatively in the proposed dataset. Given it is a historical text, the dataset might be present inappropriate biases from the past. Thus, a deeper study and investigation of the dataset should have followed. For example, a simple text-matching filtering approach with a list of blocked words would have been useful.